# Circulating MicroRNAs as Novel Potential Diagnostic Biomarkers for Osteosarcoma: A Systematic Review

**DOI:** 10.3390/biom11101432

**Published:** 2021-09-30

**Authors:** Thaís Borges Gally, Milena Magalhães Aleluia, Grasiely Faccin Borges, Carla Martins Kaneto

**Affiliations:** 1Department of Health Sciences, Universidade Estadual de Santa Cruz, llhéus 45662-900, BA, Brazil; thaibga@gmail.com; 2Department of Biological Sciences, Universidade Estadual de Santa Cruz, Ilhéus 45662-900, BA, Brazil; mmaleluia@uesc.br; 3Public Policies and Social Technologies Center, Federal University of Southern Bahia, Itabuna 45613-204, BA, Brazil; grasiely.borges@gmail.com

**Keywords:** circulating microRNAs, microRNAs, osteosarcoma, diagnostic

## Abstract

Osteosarcoma (OS) is a fast-progressing bone tumor with high incidence in children and adolescents. The main diagnostic methods for OS are imaging exams and biopsies. In spite of the several resources available for detecting the disease, establishing an early diagnosis is still difficult, resulting in worse prognosis and lower survival rates for patients with OS. The identification of novel biomarkers would be helpful, and recently, circulating microRNAs (miRNAs) have been pointed to as possible non-invasive biomarkers. In order to assess the effectiveness of miRNA research, we performed a systematic review to assess the potential role of circulating miRNAs as biomarkers for OS diagnosis. We performed a search in various databases—PubMed, LILACS (Literatura Latino-americana e do Caribe em Ciências da Saúde), VHL (Virtual Health Library), Elsevier, Web of Science, Gale Academic One File—using the terms: “Circulating microRNAs” OR “plasma microRNAs” OR “serum microRNAs” OR “blood microRNAs” OR “cell-free microRNAs” OR “exosome microRNAs” OR “extracellular vesicles microRNAs” OR “liquid biopsy” AND “osteosarcoma” AND “diagnostic”. We found 35 eligible studies that were independently identified and had had their quality assessed according to Quality Assessment of Diagnostic Accuracy Studies (QUADAS-2) guidelines. Despite the useful number of publications on this subject and the fact that several microRNAs showed excellent diagnostic performance for OS, the lack of consistency in results suggests that additional prospective studies are needed to confirm the role of circulating miRNAs as non-invasive biomarkers in OS.

## 1. Introduction

Osteosarcoma (OS), the most common type of cancer in children and adolescents, is a rapidly progressing bone tumor that is characterized by the production of osteoids by malignant cells [1]. Epidemiological data indicate that the incidence of OS is 0.2–0.3 per 100,000 cases per year, increasing to 0.8–1.1 per 100,000 cases per year in the population aged 15 to 19 [2].

Currently, diagnostic methods for OS consist of imaging examinations, such as X-rays, computed tomography and magnetic resonance imaging, and biopsy, which is considered the gold standard for diagnostic confirmation [2,3]. Although there are several resources for detecting the disease, establishing an early diagnosis is still difficult, which results in worse prognosis and lower survival rates for patients. Therefore, it would be helpful to discover new, less invasive diagnostic methods, with greater sensitivity and specificity, capable of detecting OS in early stages and with greater reliability [4]. In this context, studies involving microRNAs point to them as a good strategy to guarantee the early detection of OS, enabling better prognosis and increasing the survival of patients.

MicroRNAs are described as possible biomarkers because of their contribution to the pathogenesis of different types of cancer. They show an aberrant expression that is supposed to affect several biological processes, such as cell proliferation, differentiation and apoptosis, by controlling the expression of target genes and acting as tumor suppressors or oncogenes [5]. Circulating microRNAs, which are particularly stable in body fluids and, because of this, are proposed as excellent non-invasive cancer biomarkers, have been investigated for their diagnostic and prognostic potential [6]. These molecules are released from tissues into the circulatory system as a pathology develops. Furthermore, differential expressions of circulating microRNAs have been reported in several pathological conditions, including cancer [7,8].

Many studies on the use of circulating microRNAs as biomarkers in the diagnosis of OS have been carried out and published. They concluded that several microRNAs were found to be differentially expressed—either up- or downregulated [4,9]. Despite the useful number of publications on this subject, there are some inconsistencies between them, due to differences in the detection methods, the analysis of microRNAs expression, etc.

Considering this, we have gathered several studies to help in understanding the current research scenario on this topic through a systematic review. Systematic reviews are important because they concern important and published scientific studies, providing ample review of the state-of-the-art research on the topic, and allowing suggestions for improvements to obtain more striking results. In this context, it is imperative to develop a rigorous strategy for microRNA quantitation experiments. We also intend to suggest guidelines to avoid the potential bias introduced by differences in the amounts of starting material, sample collection and quality, RNA isolation, and PCR efficiency, since reverse transcription quantitative polymerase chain reaction (RT-qPCR), which is the most commonly used method for quantifying circulating microRNAs, could compromise the use of microRNAs as cancer biomarkers in serum or plasma samples.

## 2. Material and Methods

### 2.1. Literature Search Strategy

In accordance with predefined protocols and aiming at the identification of studies related to circulating microRNAs as diagnostic biomarkers for the detection of osteosarcoma, we performed a systematic literature search on the PubMed, LILACS (Literatura Latino-americana e do Caribe em Ciências da Saúde), VHL (Virtual Health Library), Elsevier, Web of Science, and Gale Academic One File databases for eligible articles published between 1 January 2015 and 29 April 2020. The search terms were: “Circulating microRNAs” OR “plasma microRNAs” OR “serum microRNAs” OR “blood microRNAs” OR “cell-free microRNAs” OR “exosome microRNAs” OR “extracellular vesicles microRNAs” OR “liquid biopsy” AND “osteosarcoma” AND “diagnostic”.

### 2.2. Eligibility Criteria

We considered studies eligible if they met the following criteria: (1) studies evaluating circulating microRNAs expression in human samples, comparing OS patients with healthy Control subjects; (2) studies that employed blood specimens, including serum, plasma and exosome vesicles; (3) studies that made a definitive diagnosis of OS through histopathological examination. The exclusion criteria were the following: (1) non-original articles or articles published in the form of letters to the editor, opinion pieces, reviews, editorials, case reports, expert opinions, protocols, conference or meeting abstracts, comments, or meta-analyses; (2) studies not related to circulating microRNAs’ expression or that evaluated microRNA expression only in cell lines or tissues; (3) studies with experiments on animal models only; (4) studies with insufficient or unqualified data.

### 2.3. Data Extraction and Statistical Analysis

Only articles in English, Portuguese or Spanish were included. Duplicate publications were also removed. Relevant and qualified studies were independently selected by two investigators who were also responsible for data extraction. The following data were retrieved from all included studies: basic information (first author, year of publication and country of research), patients’ characteristics (ethnicity of research population and number of participants, mean or median age, gender, histologic type and stage of the tumor), type of sample (total blood, plasma, serum or exosome vesicles), total number of cases and controls, target microRNAs or microRNA panels investigated, detection or measurement method, endogenous control used for normalization analysis, normalization method and diagnostic related parameters, such as AUC, sensitivity, specificity or expression variation. Any inconsistency was analyzed by further discussion among the authors.

### 2.4. Quality Assessment of the Included Studies

The search method and the quality of the articles were evaluated by two reviewers, and any disagreements were resolved by a third person. Quality assessment of the eligible studies was performed by two independent investigators and was conducted using QUADAS-2 (Quality Assessment Tool for Diagnostic Accuracy Studies) [10] to estimate the risk level of bias. Basically, this evaluates four components: (a) patient selection; (b) index test; (c) reference standard and (d) flow and timing. The risk level of bias is classified as “low”, “high” or “unclear” based on the answers to questions included in each component. The tool also allows for evaluating the clinical applicability, which can also be judged as “low”, “high” or “unclear”.

### 2.5. Study Registration

The retrieved studies were assessed following the criteria established by the preferred reporting items for systematic review and meta-analysis (PRISMA) guidelines for systematic review [11], with a PROSPERO registration number of CRD42020192655.

## 3. Results

### 3.1. Literature Search Results

As shown in Figure 1, via the initial literature search, we selected 282 articles. After removing 64 duplicates, the remaining 218 articles were submitted to title and abstract assessment. A further 141 articles were excluded, because they were letters, reviews, editorials, case reports, expert opinions, protocols, conference or meeting abstracts, comments or meta-analyses, or articles not related to our topic. The remaining 77 articles were submitted to full-text review, of which 42 articles were excluded because they were not realized in human samples or because they were not about circulating microRNAs. Finally, 35 studies were included in this systematic review. The characteristics of these studies are displayed in Table 1 and Table 2.

### 3.2. Main Results and Study Quality Assessment

First, we were interested in determining how microRNA candidate selection occurs. As shown in Table 1, from the 35 included articles, 9 employed some type of high-throughput analysis for microRNA candidate selection, of which 1 performed microarray analysis, 3 used sequencing and 5 employed some type of microRNA PCR panel. Regarding the types of samples evaluated in these high-throughput analyses, four articles used non-pooled samples, three used pooled samples, one used a cell lineage, and in one article, the screening analyses were conducted in non-human samples. In total, 2 articles selected microRNA candidates by analyzing public microRNA expression datasets, and the remaining 24 selected target miRNAs from the literature.

As shown in Table 2, 32 of the 35 included articles were from East Asia, 1 was from Italy, 1 from the United States and 1 from Mexico, mostly published in 2015 and 2018. The median number (range) of control and case subjects was 43.91 (3–133) and 72.70 (3–185), respectively, and the majority cases included metastatic and non-metastatic patients, except for Tian et al. [33]. Of the 35 studies, 10 did not state whether metastatic patients were included. Serum samples were the evaluated specimens in 28 articles, 5 studies analyzed plasma samples, 1 analyzed PBMC (peripheral blood mononuclear cells) and 1 evaluated microRNA expression in extracellular vesicles. Quantitative real-time PCR was used to evaluate microRNAs expressions in almost all the studies, and microRNA U6 was the most used endogenous reference microRNA for normalization purposes. However, significant differences were found in the included studies, reinforcing the need to evaluate the choice of normalization method to minimize quantitation errors and technical variability in experiments.

In total, 20 studies reported the upregulation of 38 different microRNAs in OS patients, and 16 studies reported that 22 different microRNAs were downregulated in this group of patients. Among the 35 included studies, 10 only described whether the candidate microRNA expression was up- or downregulated, and 25 described microRNAs with diagnostic potential for osteosarcoma, with a diagnostic performance of ≥0.70 AUC. Among the 16 studies that reported sensitivity and specificity, both exceeded 80% in the 3 studies of individual miRNAs and the 1 study of miRNA combinations of four microRNAs, and both exceeded 70% among the 13 individual miRNAs studies and the 1 miRNA combination study.

Several microRNAs showed excellent diagnostic performances for osteosarcoma. The AUC values of MiR-195-5p, MiR-320a and MiR-374a-5p in Lian’s et al. (2015) study [15] were 0.9029, 0.9188 and 0.9173, respectively. In Wang et al.’s (2015) study [17], the AUC value of MiR-152 was 0.956. The best values of sensitivity and specificity were also found for MiR-152 in Wang et al.’s (2015) study, and were of 96.2% and 92.5% [17].

The MiR-320a and MiR-95-3p expressions were evaluated in three different studies each. MiR-320a expression was evaluated in Monterde-Cruz et al.’s (2018) [2], Hui et al.’s (2015) [14] and Lian et al.’s (2015) [15] studies, but only in Lian et al.’s (2015) study was a differential expression observed; MiR-320a was found to be more expressed in OS samples than in control subjects, and the AUC value was 0.9188. MiR-95-3p expression was evaluated in Hui et al.’s (2015) [14], Zhao et al.’s (2018) [36] and Niu et al.’s (2016) [22] studies, but conflicting results were observed. Zhao et al. (2018) found that MiR-95-3p expression was upregulated in OS samples, and did not comment on AUC, sensitivity or specificity values, but Niu et al. (2016) described a lower expression of this microRNA, with an AUC value of 0.863.

**Table 2 biomolecules-11-01432-t002:** Summary of included studies using circulating miRNAs as biomarkers of osteosarcoma.

			Control Group			Case Group												
Author	Year	Ethnicity	N	Sex	Mean Age (y)	N	Sex	Mean Age (y)	Metast	Specim	Det Met	Normaliz.	Method for Expression Level Calculation	Differentially Expressed MicroRNAs	Up- or Downregulation Description Only	AUC	SEN	SPE
Allen-Rhoades et al. [12]	2015	American	30	N/I	18	40	2M 17F	13.41	20 yes, 19 no	plasma	qPCR	miR-320a + miR-15a- 5p + CqUniSp2)/3	2^−ΔCt^	miR-205-5p miR-214 miR-335-5p miR-574-3p	No	MiR- 205-5p: 0.70, MiR-214: 0.8, MiR-335-5p: 0.78, MiR-574-3p: 0.88	N/I	N/I
Cai et al. [13]	2015	Asian	60	N/I	N/I	166	96M 70F	<55:72 ≥55:94	42 yes, 124 no	serum	qPCR	U6	2^−^^ΔΔ^^Ct^	MiR-195	No	0.892	88.0%	83.3%
Hui et al. [14]	2015	Asian	20	12M 8F	14.3	20	13M 7F	13	2 yes, 11 no	serum	qPCR	cel-miR-39	2^−ΔΔCt^	miR-106a-5p miR-16-5p miR-20a-5p miR-25-3p miR-425-5p miR-451a miR-139-5p	No	miR-106a-5p: 0.7255 miR-16-5p: 0.7686 miR-20a-5p: 0.8471 miR-25-3p: 0.7961 miR-425-5p: 0.7765 miR-451a: 0.7961 miR-139-5p: 0.7098	N/I	N/I
Lian et al. [15]	2015	Asian	90	44M 46 F	16.2	90	43M 47F	15.8	18 yes, 72 no	plasma	qPCR	comparison of the miRNA concentrat. to the serum volume	comparison of the miRNA concentrat. to the serum volume	miR-195-5p miR-199a-3p miR-320a miR-374a-5p	No	miR-195–5p: 0.9029 miR-199a-3p: 0.9025 miR-320a: 0.9188 miR-374a-5p: 0.9173 4-miRNAs: 0.608	4-miRNAs: 91.1%	4-miRNAs: 94.4%
Tang et al. [16]	2015	Asian	60	N/I	N/I	166	96M 70F	<55:72 ≥55:94	42 yes, 124 no	serum	qPCR	U6	2^−ΔΔCt^	MiR-27a	No	0.867	70.01%	98.30%
Wang et al. [17]	2015	Asian	20	N/I	N/I	80	40M 40F	≤19:40 >19:40	12 yes, 68 no	serum	qPCR	U6	2^−ΔΔCt^	MiR-152	No	0.956	96.2%	92.5%
Wang et al. [18]	2015	Asian	20	N/I	N/I	100	66M 34F	<20:69 ≥20:31	42 yes, 58 no	serum	qPCR	U6	2^−ΔΔCt^	MiR-191	No	0.858	74.00%	100.0%
Yang et al. [19]	2015	Asian	50	N/I	N/I	108	78M 30F	<20:40 ≥20:68	40 yes, 68 no	serum	qPCR	RNU6	2^−ΔΔCt^	MiR-221	No	0.844	65.7%	100.0%
Zhou et al. [20]	2015	Asian	60	38M 22F	≥20:23 <20:37	60	38M 22F	≥20:23 <20:37	8 yes, 52 no	serum	qPCR	comparison of the miRNA concentrat. to the serum volume	comparison of the miRNA concentrat. to the serum volume	MiR-199a-5p	No	0.8606	88.33%	76.67%
Cao et al. [21]	2016	Asian	20	N/I	N/I	60	32M 28F	≤18:37 >18:23	9 yes, 51 no	serum	qPCR	RNU48	2^−ΔΔCt^	MiR-326	No	0.897	83.7%	94.5%
Li et al. [3]	2016	Asian	46	27M 19F	19.6	46	27M 19F	19.6	N/I	serum	qPCR	U6	2^−ΔΔCt^	MiR-17	Yes	N/I	N/I	N/I
Niu et al. [22]	2016	Asian	133	71M 62F	≤15:59 >15:74	133	71M 62F	≤15:59 >15:74	68 yes, 65 no	serum	qPCR	U6	2^−^^ΔΔ^^Ct^	MiR-95-3p	No	0.863	N/I	N/I
Pang et al. [23]	2016	Asian	130	N/I	N/I	185	110M 75F	<55:73 ≥55:112	57 yes, 128 no	serum	qPCR	U6	2^−ΔCt^	MiR-497	No	0.848	N/I	N/I
Sun et al. [24]	2016	Asian	62	N/I	N/I	62	N/I	N/I	N/I	serum	qPCR	U6	2^−ΔΔCt^	MiR-24	Yes	N/C	N/C	N/C
Zhou et al. [25]	2016	Asian	40	N/I	N/I	40	25M 15F	≥15:27 <15:13	N/I	serum	qPCR	U6	2^−ΔΔCt^	MiR-421	Yes	N/C	N/C	N/C
Fujiwara et al. [26]	2017	Asian	8	4M 4F	N/I	14	7M 7F	0–10:2 11–20:8 ≥21:4	1 yes, 13 no	serum	qPCR	N/I	2^−^^ΔΔ^^Ct^	miR-25-3p miR-17-3p	No	MiR-25-3p: 0.868 MiR-17-3p: 0.720	MiR-25-3p: 71.4% MiR-17-3p: 64.3%	MiR-25-3p: 92.3%; MiR-17-3p: 84.6%
Liu et al. [27]	2017	Asian	10	N/I	N/I	20	N/I	N/I	N/I	serum	qPCR	N/I	N/I	MiR-598	Yes	N/C	N/C	N/C
Wang et al. [28]	2017	Asian	20	8M 12F	24.5	102	54M 48F	Low: 17.3 High: 16.4	36 yes, 66 no	serum	qPCR	RNU6B	N/I	MiR-491	Yes	N/I	N/I	N/I
Xie et al. [29]	2017	Asian	3	N/I	N/I	3	N/I	N/I	N/I	PBMC	qPCR	U6	2^−^^Δ^^Ct^	hsa-miR-221-5p hsa-miR-26b-5p hsa-miR-21-5p hsamiR-5706 hsa-miR-656-3p	Yes	N/C	N/C	N/C
Cong et al. [30]	2018	Asian	50	N/I	N/I	114	62M 52F	≥18: 71 <18: 43	60 yes, 54 no	serum	qPCR	RNU6	2^−^^ΔΔ^^Ct^	MiR-124	No	0.846	79.8%	86%
Li, Song et al. [31]	2018	Asian	76	N/I	N/I	76	N/I	N/I	N/I	plasma	qPCR	U6	2^−ΔΔCt^	MiR-542-3p	No	0.841	77.8%	93.6%
Liu, Zhao et al. [32]	2018	Asian	95	N/I	N/I	95	63M 32F	<20: 69 ≥20: 26	37 yes, 58 no	serum	qPCR	U6	2^−ΔΔCt^	MiR-375	No	0.89	82.1%	74.7%
Monterde-Cruz et al. [2]	2018	Mexican	15	9M 6F	20	15	9M6F	20	13 yes, 2 no	serum	qPCR	RNU6	2^−ΔΔCt^	miR-215-5p miR-642a-5p	No	miR-215-5p: 0.8667, miR-642a-5p: 0.8413, 2-miRNAs: 0.8520	N/I	N/I
Tian et al. [33]	2018	Asian	30	N/I	N/I	65	35M 30F	≤12:35 >12:30	No	serum	qPCR	U6	N/I	MiR-337-5p	No	0.7761	N/I	N/I
Xu et al. [34]	2018	Asian	30	N/I	N/I	30	N/I	N/I	N/I	serum	qPCR	U6	N/I	MiR-411	Yes	N/C	N/C	N/C
Yao et al. [35]	2018	Asian	70	N/I	N/I	152	8M 65F	<55: 84 ≥55: 68	21 yes, 131 no	serum	qPCR	U6	2^−ΔΔCt^	MiR-101	No	0.850	78.95%	82.86%
Zhao et al. [36]	2018	Asian	N/I	N/I	N/I	N/I	N/I	N/I	N/I	serum	qPCR	N/I	N/I	MiR-95-3p	Yes	N/C	N/C	N/C
Zhou et al. [37]	2018	Asian	50	N/I	N/I	98	62M 36F	<19: 47 ≥19: 51	30 yes, 68 no	serum	qPCR	cel-MiR-39	2^−ΔΔCt^	MiR-139-5p	No	0.846	76.5%	80%
Zhou et al. [38]	2018	Asian	7	4M 3F	N/I	7	4M 3F	N/I	N/I	serum	qPCR	U6	2^−ΔΔCt^	MiR-22	Yes	N/C	N/C	N/C
Cuscino et al. [9]	2019	Italian	3	N/I	N/I	5	M	16.8	2 yes, 3 no	plasma	Digital PCR	U6	2^−ΔCt^	5 new microRNA candidates	Yes	N/I	N/I	N/I
Huang et al. [4]	2019	Asian	30	22M 28F	≤16: 26 >16: 24	50	22M 28F	≤16: 26 >16: 24	18 yes, 32 no	serum	qPCR	U6 and cel-MiR-39	ΔCt = CtmiRNA− CtmiR—39/U6	MiR-487-a MiR-493-5p MiR-501-3p MiR-502-5p	No	miR-487a: 0.83, miR-493-5p: 0.79, miR-501-3p: 0.82, miR-502-5p: 0.83, 4-miRNAs: 0.89	N/I	N/I
Huang, Sun et al. [39]	2019	Asian	50	32M 18F	≤14: 30 >14: 20	50	3M 14F	≤14:31 >14: 19	11 yes, 39 no	plasma	qPCR	U6, cel-MiR-39	ΔCt = CtmiRNA− CtmiR—39/U6; ΔCtCt = ΔCtpatient− Ctcontrol	MiR-663a	No	0.86	67.35%	89.8%
Zhu et al. [1]	2019	Asian	25	N/I	N/I	55	N/I	N/I	N/I	serum	qPCR	GAPDH	2^−ΔΔCt^	hsa_circ_0000885	No	0.783	N/I	N/I
Shi et al. [40]	2020	Asian	60	N/I	N/I	124	79H e 45 M	<50:72 ≥50:52	33 yes, 91 no	serum	qPCR	cel-miR-39	2^−ΔΔCt^	MiR-194	No	0.855	84.2%	79.1%
Zhang et al. [41]	2020	Asian	20	12M 8F	18.5	41	27M 14F	16	14 yes, 27 no	Extrac. Vesicul.	qPCR	U6, Cel-mir-39 e let-7i-5p	2^−ΔΔCt^	MiR-101	No	0.7957	N/I	N/I

Figure 2 shows the QUADAS-2 quality evaluation results. The results indicate that the included studies had low to moderate scores because they had unclear information about patients and the reference standard selection, as well as low applicability concerns.

## 4. Discussion

Although perioperative management, surgery and multiagent chemotherapy have greatly evolved in recent years, OS is still the most common malignant bone tumor in children and adolescents [42], with an incidence rate of 4.5/million/year [43], and with a very high morbidity and mortality rate [44]. Besides this, in recent years, no great progressions in OS diagnosis or early detection have been accomplished for clinical application, despite efforts to identify more tumor-related regulators and molecules involved in the growth and metastasis of this tumor [45,46].

MicroRNAs are described as small non-coding RNAs, composed of 22 to 24 nucleotides, which regulate gene expression in several cellular processes. The deregulated expressions of these microRNAs interfere with the cell cycle, potentially causing abnormal cells. This deregulation is closely associated with the development of several pathologies, including cancer [3]. MicroRNAs can be found not only in cells and tissues, but also circulating freely in body fluids, such as serum, plasma and urine, among others. These circulating microRNAs have remarkable stability in body fluids, allowing us, by analyzing their concentrations and compositions, to diagnose diseases, including OS. Thus, circulating microRNAs can act as potential biomarkers in the diagnosis of OS, being minimally invasive and effective in the early detection of the disease [20].

In this context, it is imperative to develop a rigorous strategy for microRNA quantitation experiments to avoid the potential biases introduced by differences in the amounts of starting material, sample collection and quality, RNA isolation, and PCR efficiency. These issues are especially relevant for the reverse transcription quantitative polymerase chain reaction (RT-qPCR) method, which is most commonly used to quantify circulating microRNAs. The use of microRNAs as cancer biomarkers in serum or plasma samples might be compromised if these issues are not taken into consideration.

In this systematic review, we identified a total number of 60 microRNAs from 35 studies evaluating the diagnostic potential of circulating microRNAs for osteosarcoma detection. Interestingly, a large number of studies selected microRNA candidates for expression evaluation by analyzing similar studies involving different types of cancer from the literature. However, their results might be compromised for the reasons mentioned above, and there is often limited overlap between them due to the varied sample sources or analysis means. In this context, miRNA signatures that consist of a variety of different miRNAs provided by large-scale studies are still missing, and would help to provide additional important information and to improve differentiation between pathologies, given that some microRNAs, such as MiR-21 and MiR-20a, are frequently not disease-specific.

Despite the considerable number or studies, the available data are not sufficient for a specific microRNA or group of microRNAs to be established as an OS diagnostic biomarker. First, the sample size was found to be very small in a large number of the included studies, and most of them also did not offer important information about the characterization of the OS and control samples evaluated. The description of the participant’s age, sex and other characteristics, including tumor location, subtype and the presence of metastasis, are important when clarifying to whom the study findings are applicable, allowing them to be generalized or showing their limitations. It would be helpful if the studies included broader population samples from different ethnic groups, especially from high-OS incidence countries, in order to investigate if microRNA expression is ancestry-specific. The inclusion of metastatic patients should also be re-evaluated. While only 1 study did not enroll patients with metastasis, 10 out of the 35 studies did not mention whether metastatic patients were included. Studies performing separate analyses of patients with or without metastasis could help to reveal both metastasis-specific and non-metastasis-specific miRNA biomarkers.

Chemotherapy and surgery may also affect the expression of circulating microRNAs [4,20,35] once antineoplastic drugs, for example, are demonstrated to regulate cell proliferation and angiogenesis, which may have a big impact on microRNAs expression profiling. So, in order to avoid the effect of treatment on microRNA expression, only pre-therapy OS samples should be included. In 14 of the studies included, however, this information was not available, and patients undergoing chemotherapy treatment were included in 6 of them.

Differences in the execution of methods applied to identify differentially expressed circulating microRNAs can also influence the screening of circulating microRNAs for OS detection, and thus require attention. Specimen types and their preservation [47,48], and microRNA isolation protocols [49], are some examples. MicroRNAs’ expressions and the final quantitative results can also be highly influenced by different normalization methods [50,51]. Quantitative PCR methods of microRNA expression are not currently universal, and there is no consensus about the ideal endogenous reference gene to be used for the normalization of microRNA expression data from patients with OS and other types of cancer. The most used endogenous reference genes in the included studies are RNU6B, cel-miR-39 and U6 snRNA, but other unusual microRNAs, such as MiR-320a, MiR-15a-5p and let-7i-5p, were also found. It is important to reinforce that the expressions of microRNAs used as the endogenous reference should be consistent among all samples and groups, as they play an instrumental role in the evaluation of circulating microRNA expression; as such, the development of a rigorous normalization strategy should be considered to avoid measurement errors.

Although several microRNAs showed excellent diagnostic performances for osteosarcoma, and a large number of microRNAs had their expressions evaluated, the overlapping rates of OS-specific circulating microRNAs were low in the analyzed literature. Only two microRNAs, MiR-320a and MiR-95-3p, were evaluated in three different studies each, and their expressions in different studies were sometimes inconsistent [22,36]. Consequently, future studies should perhaps focus on combining the different microRNA markers already evaluated in a diagnostic model for the early detection of OS, in addition to the identification of more circulating microRNAs, thus enhancing diagnostic power.

## 5. Conclusions

In conclusion, this systematic review suggests that, although circulating microRNAs hold great potential to be used as diagnostic markers for OS, future studies should consider a more stringent standardization of sample characterization and microRNA quantitation protocols. Verification of these OS-specific microRNAs in large-scale screening studies would also be helpful to determine their diagnostic efficiency for the early detection of OS. Our study highlights that it is imperative to develop a rigorous strategy for microRNA quantitation experiments that allows the use of microRNAs as cancer biomarkers in serum or plasma samples.

## Figures and Tables

**Figure 1 biomolecules-11-01432-f001:**
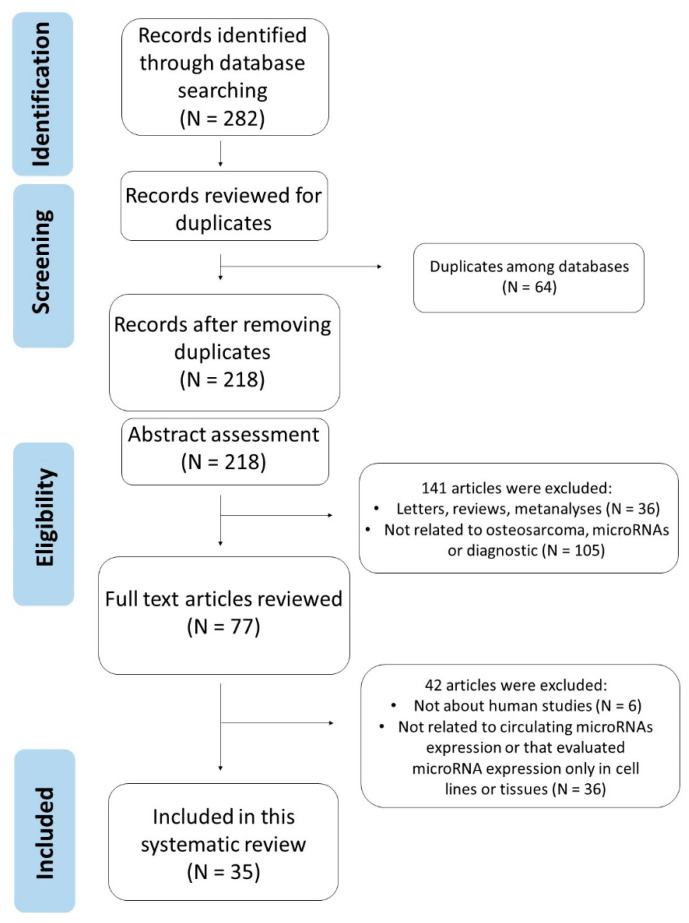
Overview of the literature search and selection process.

**Figure 2 biomolecules-11-01432-f002:**
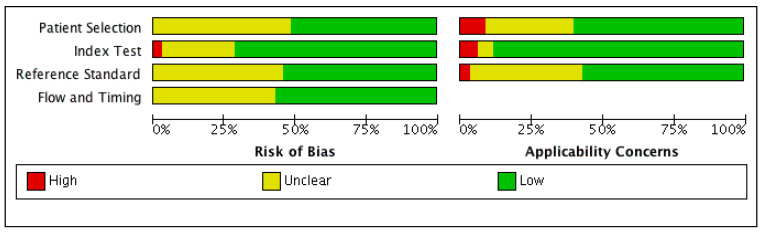
Risk of bias assessment using the QUADAS-2 tool.

**Table 1 biomolecules-11-01432-t001:** MicroRNA candidate selection methods found in the included articles.

Author	Year	Type of Global MicroRNA Expression Profiling	Were Samples Pooled?	If Yes, What Was the Number of Samples Per Pool?	If Samples Were Not Pooled, How Many Samples Per Group Were Analyzed in Large-Scale Analysis?	Were Candidate MicroRNAs Selected by Analysis of Public MicroRNAs Datasets?	Were Candidate MicroRNAs Selected by Literature Review?
Allen-Rhoades et al. [12]	2015	MicroRNA PCR panel	Conducted using non-human samples	N/A	N/A	No	No
Cai et al. [13]	2015	N/A	N/A	N/A	N/A	No	Yes
Hui et al. [14]	2015	MicroRNA PCR panel	No	N/A	3 per group	No	No
Lian et al. [15]	2015	MicroRNA PCR panel	Yes	2 pools with 10 samples each	N/A	No	Yes
Tang et al. [16]	2015	N/A	N/A	N/A	N/A	No	Yes
Wang et al. [17]	2015	N/A	N/A	N/A	N/A	No	Yes
Wang et al. [18]	2015	N/A	N/A	N/A	N/A	No	Yes
Yang et al. [19]	2015	N/A	N/A	N/A	N/A	No	Yes
Zhou et al. [20]	2015	MicroRNA PCR panel	Yes	3 pools with 10 samples each	N/A	No	No
Cao et al. [21]	2016	N/A	N/A	N/A	N/A	No	Yes
Li et al. [3]	2016	N/A	N/A	N/A	N/A	No	Yes
Niu et al. [22]	2016	N/A	N/A	N/A	N/A	No	Yes
Pang et al. [23]	2016	N/A	N/A	N/A	N/A	No	Yes
Sun et al. [24]	2016	N/A	N/A	N/A	N/A	No	Yes
Zhou et al. [25]	2016	N/A	N/A	N/A	N/A	No	Yes
Fujiwara et al. [26]	2017	Microarray	No	N/A	10 per group	No	No
Liu et al. [27]	2017	N/A	N/A	N/A	N/A	No	Yes
Wang et al. [28]	2017	N/A	N/A	N/A	N/A	No	Yes
Xie et al. [29]	2017	Sequencing	No	N/A	3 OS and 10 control subjects	No	No
Cong et al. [30]	2018	N/A	N/A	N/A	N/A	No	Yes
Li, Song et al. [31]	2018	N/A	N/A	N/A	N/A	No	Yes
Liu, Zhao et al. [32]	2018	N/A	N/A	N/A	N/A	No	No
Monterde-Cruz et al. [2]	2018	MicroRNA PCR panel	Yes	4 pools with 5 samples each	N/A	No	No
Tian et al. [33]	2018	N/A	N/A	N/A	N/A	Yes	No
Xu et al. [34]	2018	N/A	N/A	N/A	N/A	No	Yes
Yao et al. [35]	2018	N/A	N/A	N/A	N/A	No	Yes
Zhao et al. [36]	2018	N/A	N/A	N/A	N/A	No	Yes
Zhou et al. [37]	2018	N/A	N/A	N/A	N/A	No	Yes
Zhou et al. [38]	2018	N/A	N/A	N/A	N/A	No	Yes
Cuscino et al. [9]	2019	Sequencing	Conducted using cell lineages samples	N/A	N/A	No	No
Huang et al. [4]	2019	N/A	N/A	N/A	N/A	Yes	No
Huang, Sun et al. [39]	2019	N/A	N/A	N/A	N/A	No	No
Zhu et al. [1]	2019	N/A	N/A	N/A	N/A	No	Yes
Shi et al. [40]	2020	N/A	N/A	N/A	N/A	No	Yes
Zhang et al. [41]	2020	Sequencing	No	N/A	1 per group	No	Yes

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
