# Peer review of "Circulating MicroRNAs as Novel Potential Diagnostic Biomarkers for Osteosarcoma: A Systematic Review"

_biomolecules, 2021, doi:10.3390/biom11101432_

Round 1
Reviewer 1 Report
**COMMENTS TO THE AUTHORS**
This manuscript by Gally et. al. aims to systematically review potential for microRNAs as circulating biomarkers of osteosarcoma (OS). By literature search authors describe largely discrepant and study-specific pools of microRNAs with the potential for acting as biomarkers of OS.
The literature search design is sound and execution of analysis and quality of data do not raise major concerns. Authors bring a valid point for need of more streamlined and benchmarked protocols for detection of miRNAs in the specimens from OS patients. However, given largely different ethnicities of the patient cohorts, different types of material and variety of miRNA detection methods, Authors fall short of providing comprehensive overview of possible reasons for observed discrepancies. Therefore, it remains difficult to interpret some of the observations and additional analysis is necessary to warrant the Authors’ conclusions.
Specific points
* Main concern is that Authors mainly ascribe the source of variability between different studies to normalization controls. Although this is a valid point, it seems unlikely that it would substantially contribute to the variability of the data given that most of the analyzed studies employed either U6 or cel-miR-39 controls. These are widely used for this type of studies. Instead it is likely that other factors, that Authors do not discuss thoroughly, especially targeted approach of vast majority of studies (27 out of 35) would obscure identification of high confidence miRNA hits. Authors may consider including additional analysis focused only on 9 high-throughput, unbiased studies. Results should be discussed taking into account differences in biological material, ethnicity or number of patients enrolled. That would facilitate interpretation of the data.
* Authors raise a point about different inclusion criteria for metastatic patients in reported studies. If the analysis was performed separately on studies with or without inclusion of the metastatic patients it could reveal both metastasis- as well as pan-OS-specific miRNA biomarkers. If possible, Authors should include this analysis in their manuscript.
* Authors discuss discrepancy observed for the miR-95-3p that was identified in more than one study with different predictive power. However, mentioned studies of Zhao et al. (2018) and Niu et al. (2016) asses largely different experimental settings, with Zhao and colleagues looking at the cellular context with cell lines as model system, whilst Niu et al. validate solely the clinical predictive power. Description of these discrepancies could be included in the text. Also, Zhao et al. do not describe predictive power of miR-95-3p for survival of OS patients, therefore Authors statement that in this study miR-95-3p did not inform AUC is incorrect.
Editing comments
* what is the line of reasoning behind excluding articles published as letters?
*in the section describing distribution of the 35 articles in terms of approach for detection of miRNAs as biomarkers of OS numbers do not add up to 35 (9 high-throughput, 2 with public miRNA expression databases and 25 with candidate approach).
* not all of the literature is cited – it is already apparent from the count of the cited papers (28) as compared to the studies included in the systematic review (35).
* There are multiple spelling mistakes in the text, e. g. several instances of using “de” instead of “the”, e. g. on the page 2. Furthermore, manuscript text would benefit from thorough re-reading by native English speaker.

Reviewer 2 Report
I think that the limitation is don't have studies with large cohorts of patients required to validate the performance of circulating ncRNAs as independent and reproducible diagnostic biomarkers.
Reviewer 3 Report
Extensive review on the available data on microRnas in Osteosarcoma. The literature is well analyzed and the data clearly presented. The major interest is the highlight of the difficulties to draw any conclusion despite the number of published papers (35 selected publication) because of the lack of homogeneity and standardization between studies and usually the small number of samples collected . It is interesting for the readers and the field in general to realize that without harmonized methodology it will be very difficult to use the laboratory data in clinic. This analyze could probably be generalized to several publications on microRNas as biomarker in different disease or clinical situation.
So even there is no positive conclusion on a set of useful microRnas as biomarkers, I think the review should be accepted for publication. In their conclusion the authors could stress even more the need of technical workshop about standardization or consortium to join effort on microRnas biomarkers, if we don't want to waste money in repeating small local study not having the power to change patients clinical management or to write new guidelines.

Round 2
Reviewer 1 Report
Authors considerably improved the quality of the manuscript, adressing all the points raised in the revision.